# Is the outcome of fitting hearing aids to adults affected by whether an audiogram-based prescription formula is individually applied? A systematic review protocol

Ibrahim Almufarrij [1,2] Harvey Dillon [1,3] Kevin J Munro [1,4]

¹Manchester Centre for Audiology and Deafness, School of Health Sciences, The University of Manchester, Manchester, UK
²Department of Rehabilitation Sciences, College of Applied Medical Sciences, King Saud University, Riyadh, Saudi Arabia
³Department of Linguistics, Macquarie University, Sydney, New South Wales, Australia
⁴Manchester University Hospitals NHS Foundation, Trust Manchester Academic Health Science Centre, Manchester, UK

**Correspondence to**
Mr Ibrahim Almufarrij;
ialmufarrij@ksu.edu.sa

## ABSTRACT

**Introduction** Hearing aids are typically programmed using the individual's audiometric thresholds. Developments in technology have resulted in a new category of direct-to-consumer devices, which are not programmed using the individual's audiometric thresholds. This review aims to identify whether programming hearing aids using the individual's audiogram-based prescription results in better outcomes for adults with hearing loss.

**Methods and analysis** The methods of this review are reported in line with Preferred Reporting Items for Systematic Reviews and Meta-Analyses Protocols guidelines. On 23 August 2020, eight different databases were systematically searched without any restrictions: EMBASE, MEDLINE, PubMed, PsycINFO, Web of Science, Cochrane Library, Emcare and Academic Search Premier. To ensure that this review includes the most recent evidence, the searches will be repeated at the final write-up stage. The population of interest of this review will be adults with any degree or type of hearing loss. The studies should compare hearing aids programmed using an audiogram-based prescription (and verified in the real ear) with those not programmed on the basis of the individual's audiogram. The primary outcome of interest is consumers' listening preferences. Hearing-specific health-related quality of life, self-reported listening ability, speech intelligibility of words and sentences in quiet and noisy situations, sound quality ratings and adverse events are the secondary outcomes of interest. Both randomised and non-randomised controlled trials will be included. The quality of each individual study and the overall evidence will be assessed using Downs and Black's checklist and the Grading of Recommendations, Assessment, Development and Evaluations tool, respectively.

**Ethics and dissemination** We will only retrieve and analyse data from published studies, so no ethical approval is required. The review findings will be published in a peer-reviewed journal and presented at scientific conferences.

**PROSPERO registration number** CRD42020197232.

## Strengths and limitations of this study

► The primary outcome is listening preference, measured in a within-group (cross-over) experiment, but both within-group and between-group design studies will be included as the latter design can contribute to secondary outcomes.
► Amplification devices preprogrammed for generic mild or moderate hearing loss will be included among the non-individually programmed devices.
► We will only include studies that compare programming approaches using the same hearing device.
► Magazine articles, conference abstracts, clinical guidelines and theses will be excluded.
► The results will help decision-makers and hearing health professionals to provide better patient-centred care.

worldwide.[1] It is also the most common cause for years lived with disability.[2] Hearing loss can directly affect people's ability to communicate, which can consequently lead to depression and poor social interaction.[3] It is also associated with a reduced quality of life and an increased risk of dementia.[4] Hearing loss can be partially and successfully mitigated through the use of complex electroacoustic amplification devices known as hearing aids.[5]

Hearing aids are the primary and the most common clinical intervention for hearing loss because they are very effective in improving hearing-related quality of life.[6 7] The ultimate goals for hearing aids are to restore the audibility of soft level sounds, maximise the intelligibility of conversational-level sounds and ensure that loud sounds are within comfortable levels.[8]

In the early 1940s and 1950s, the most controversial aspects of clinical audiology were hearing aid selection and fitting, as experts could not agree on the most

## INTRODUCTION

Hearing loss is a debilitating health condition that affects more than 450 million individuals

appropriate approaches.[9] Since this period, prescriptive approaches have evolved to provide recommended gains for each audiometric frequency and input level. The concept of hearing aid prescriptions goes back to the early 1940s when Jones and Knudsen developed the audiogram mirroring prescription procedure, in which every decibel (dB) of loss was compensated with an additional dB of gain.[10] The prescription was then developed by Watson and Knudsen[11] to incorporate the individual's most comfortable level.[11] Shortly after, Lybarger[12] developed the half-gain rule, in which every additional 2 dB of hearing loss was compensated with an additional 1 dB of gain.[12] Since then, many other prescription procedures have been developed, but very few of them have been comprehensively studied and validated. The invaluable contribution of Denis Byrne (who is known as the father of prescriptive hearing aid fitting) to the development of prescription procedures has led to the indubitable universal acceptance of hearing aid prescriptions.[13]

At present, several acoustic laboratories worldwide have developed validated audiogram-based prescription formulae, such as National Acoustic Laboratories Non-Linear (NAL-NL1 and NAL-NL2)[14 15] and Desired Sensation Level.[16] These prescription formulae aimed to provide the optimal amount of amplification for each audiometric frequency.[17] Such prescriptions have been revised and updated multiple times based on a combination of theoretical derivation and empirical data.[18]

## Rationale

Numerous direct-to-consumer hearing devices have been mass produced and marketed to customers. Although these hearing devices vary considerably in quality, a few have comparable electroacoustics with conventional hearing aids that are programmed and fitted by audiologists.[19] Moreover, some of these hearing devices allow consumers to use their smartphone or remote controls to adjust the amplification characteristics according to their preferences.[20] Allowing users to take control of their amplification characteristics may help them to achieve a response that is better than one based on prescription targets that reflect what is optimal for an average person who has the same audiogram as that individual user. Alternatively, imprecision in adjustments made by the individual, or an inadequately adjustable hearing aid, may result in poorer outcomes than for hearing aids adjusted by a clinician on the basis of the individual's audiogram. In addition, individuals with hearing loss may prefer different amplification characteristics for different acoustic environments,[21 22] in ways that vary beyond the variation prescribed by existing non-linear prescription formula.

## Objectives

It is unknown whether the use of individually prescribed amplification characteristics, using audiogram-based prescription formulae, and verified using real-ear probe microphone measurements, provide better outcomes than using amplification characteristics that are not prescribed on the basis of the individual's audiogram (eg, users adjust the amplification characteristics of their device based on their listening needs). Mueller systematically reviewed studies that compared audiogram-based prescriptions with each other or with the same prescription but with the user's adjustment to the overall gain.[23] Direct-to-consumer hearing devices that allow users to fine-tune the amplification characteristics without audiometric thresholds were not widely available at the time of Mueller's review. Thus, this review aims to investigate whether the outcomes for adults are better when hearing aids are programmed using an audiogram-based prescription formulae.

## METHOD AND ANALYSIS

This review protocol was preregistered in International Prospective Register of Systematic Reviews. The method of this systematic review is reported in line with the Preferred Reporting Items for Systematic Review and Meta-Analyses (PRISMA-P) guidelines.[24]

## Eligibility criteria

The eligibility criteria of this review are specified and described in line with the participants, interventions, comparators, outcomes and study designs criteria.

## Participants

Adults (≥18 years) with any defined degree and type of hearing loss will be included. If only qualitative descriptions of age and hearing thresholds are reported in a trial, then the study will still be included. Studies that include both adults and children will be included if the results were separately analysed and reported. In addition, studies that include participants with normal hearing and hearing loss will not be included unless they were analysed independently.

## Interventions

The intervention of interest is any amplification device (ie, conventional hearing aids, direct-to-consumer hearing devices or simulated hearing aids) that has not been programmed (using either a verified or a manufacturer proprietary prescription formula) using the individual user's hearing thresholds. Amplification devices preprogrammed for generic mild or moderate hearing loss will be included, unless the experimental protocol tested the device only on people selected because they have audiograms that closely matched the generic hearing loss target(s).

The initial plan was to exclude studies that used simulated hearing aids because they are less likely to reflect real-life benefit. However, to maximise the number of included studies, we broadened the inclusion criteria to include simulated hearing aids.

## Comparators

Comparators will include any amplification device (ie, conventional hearing aids, direct-to-consumer hearing

devices or simulated hearing aids) programmed using an audiogram-based prescription target (eg, NAL-NL2) and verified using a real-ear measurement system. Implantable devices, assistive listening devices and bone conduction hearing devices will be excluded. Studies using different hearing devices for the intervention and comparator will be excluded, as differences in technology, features and appearance could serve as serious confounding variables.

## Outcomes

The primary outcome of this review is the participant's listening preference. Secondary outcomes will include hearing-specific health-related quality of life (eg, Hearing Handicap Inventory for the Elderly[25]), self-reported listening ability (eg, The Speech, Spatial and Qualities of Hearing Scale[26]), speech intelligibility of words and sentences in quiet and noisy situations, sound quality ratings and adverse events (eg, discomfort, hearing aid rejection and noise-induced hearing loss). Studies reporting any of the above outcomes will be included. Studies that only quantify the deviation from the prescriptive targets will be excluded. Studies that used only predicted speech intelligibility (speech intelligibility index) as an outcome will be also be excluded.

## Study designs

We will include both randomised and non-randomised controlled trials, and cross-over designs. Conference abstracts, book chapters and theses will be excluded. Case reports, reviews and clinical guidelines will also be excluded.

## Information sources

A systematic search strategy was conducted to identify published, concluded but unpublished and ongoing experiments. The following databases were searched: EMBASE, MEDLINE, PubMed, PsycINFO, Web of Science, Cochrane Library, Emcare and Academic Search Premier. All databases were searched on 23 August 2020 with no search restrictions in relation to the publication's year, status and language. To ensure that this review includes the most recent evidence, the searches will be repeated at the final write-up stage. The initial plan was to include grey literature in the information sources, but they were excluded because a preliminary search produced no relevant records, and there is no agreed method of systematically searching such literature.

The reference lists of the included studies will be scanned to identify other relevant studies. We will also track the citation of the included studies using Google Scholar to identify additional relevant articles.

## Search strategy

A medical information specialist developed the search protocol. The search terms were developed based on free text, expert opinions and controlled terms (eg, Medical Subject Headings). The search protocol for each of the included databases is reported in online supplemental appendix 1.

## Study records
### Data management

The search result records will be extracted to EndNote V.X9 Reference Management software (Clarivate Analytics, 2018) to remove duplicated records. Study details (ie, authors, publication year, titles and abstracts) will then be transformed into a Microsoft Excel (2016) spreadsheet so that they can be easily assessed for inclusion.

## Selection process

Two of the review team will independently assess the title and abstract of all retrieved studies to determine their eligibility for inclusion. The reason for excluding any article will be documented. Disagreements will be resolved by discussion or by consulting the third author. Articles that meet the inclusion criteria will be retrieved for a full inspection. The first and second authors will then screen the retrieved records against the eligibility criteria and disagreements will be resolved by discussion with the third author. The authors of all relevant identified protocols will be asked about the publication status of their clinical experiments to identify and include all studies published up until the review is completed. The study selection process along with the main reasons for exclusion will be illustrated in a PRISMA flow diagram.

## Data collection process and data items

The first author will extract the data from all eligible studies. Another member of the review team will independently extract a small proportion of data to check for consistency. Any disagreements will be resolved by arbitration or by consulting the third author. An adapted version of the Cochrane Handbook data extraction form will be used to extract the data from the included studies. The main items of the data extraction are summarised in online supplemental appendix 2. When necessary, an extraction tool (eg, WebPlotDigitizer) will be used to extract data that are only available in figures and graphs.

## Risk of bias in individual studies

The methodological quality of the studies will be assessed using the Downs and Black (1998) checklist because it is easy to use and has acceptable validity and reliability. This tool provides numerical scores for each of the following domains: study quality, external validity, study bias, confounding and selection bias as well as power of the study. In this review, the score for the power domain (ie, question number 27) will be reduced from three points to one point because there is a dearth of knowledge about the clinically important differences in hearing aid outcomes. The full score for this domain will be awarded if sample size calculation was made and a score of zero will be awarded if it was not. However, if no power analysis was performed but the sample size is commensurate with the other studies for which one was performed, the study will not be penalised. Thus, the maximum possible score will be 28 and the quality of each study will be categorised

as excellent (26–28), good (20–25), fair (15–19) or poor quality (<14). The assessment of the risk of bias will be carried out independently by all authors. Disagreements will be resolved by discussion or decision by majority.

## Data synthesis
The data will be synthesised into a meta-analysis where possible. For between-group studies, the mean difference (when the same continuous outcomes are used across studies) or the standardised mean difference (when different continuous outcomes are used) will be calculated along with their 95% CI. The risk ratio and its 95% CI will be calculated for dichotomous outcomes. For cross-over design studies, the effect size and its 95% CI will be calculated in accordance with Cochrane Handbook recommendations, which usually require the calculation of the correlation coefficient between the evaluated interventions when the outcome is continuous variable. For the primary outcome of interest, the participants' listening preference, however, the difference in proportion between those preferring each condition and its 95% CI will be computed. A fixed-effect meta-analysis will be computed whenever the statistical heterogeneity is low; otherwise, the data will be synthesised using a random-effect meta-analysis. The generic inverse of variance approach will be used to weight each study and hence compute the effect estimate and its 95% CI. The data for each meta-analysis will be presented in a forest plot.

Asymmetrical distribution of outcomes will be assessed where either the lowest or the highest possible values of a used scale are available. The assessment involves subtracting the lowest possible value from the mean (or vice versa with the highest possible value) and dividing this by the SD. When the observed ratio is less than 1, there is strong evidence of a skewed distribution. A ratio between 1 and 2 also suggests a skewed distribution. Where possible, appropriate transformation methods will be used to transform and normalise the distribution of skewed data.

Data from studies that have different designs will be combined only if their effect sizes can be transferred to the same metric (eg, standardised mean difference) and their effect sizes estimate the same treatment effect.[27]

The data will be quantitatively instead of qualitatively synthesised when a meta-analysis is considered to be inappropriate.

## Assessment of reporting bias
Publication bias will be assessed, when possible, using a funnel plot of the studies' standard error or precision as a function of effect estimates.

## Assessment of heterogeneity
The variability across studies (for each outcome) will be assessed using the $I^2$ statistic. Percentages of 0%–40%, 41%–60% or 61%–100%, respectively, indicate low, medium or high heterogeneity. In addition, a $X^2$ test will

be performed to assess the statistical significance of the heterogeneity.

## Dealing with missing data
Missing data will be obtained by contacting the corresponding authors, where possible. Missing SDs and correlation coefficients will be estimated or imputed from the other available data using protocols explained in the Cochrane Handbook.

## Subgroup analysis
Plausible sources of heterogeneity will be explored using a subgroup analysis of the participant's age, degree of hearing loss, level of experience with hearing aids and the length of their acclimatisation period with the hearing aid.

## Confidence in the cumulative estimate
The Grading of Recommendations, Assessment, Development and Evaluations tool (GRADE) will be used to rate the overall evidence of each outcome. The assigned rating (ie, high, moderate, low or very low) reflects our confidence of the pooled estimate, that is, high-quality evidence implies that the pooled estimates are probably very close to the true effect of the intervention. Conversely, very low-quality evidence implies that the pooled estimate is likely to be substantially different from the true effect and further research is needed to strengthen our confidence in the pooled effect. The initial quality ratings for randomised and non-randomised trials will be, respectively, high and low. Cross-over designs, where each participant acts as their own control, will also initially be assigned a high quality provided cross-over order is counterbalanced across participants. The assigned rating can be upgraded or downgraded by one or two points based on the seriousness of multiple factors. Notably, three factors may upgrade the quality of evidence: (1) large effect size, (2) dose–response gradient (eg, the magnitude of the participants' preference for the audiogram-based approach (comparator) increased when the deviation from the prescribed target decreased) and (3) plausible confounding factors that likely have reduced the effect observed (eg, participants preferred the amplification characteristics for audiogram-based approach (comparator) over the other approaches (intervention) even when more advanced features were exclusively activated with the intervention). Meanwhile, study limitations, inconsistency, indirectness, imprecision and publication bias are factors that may reduce our confidence in the pooled estimates. The overall quality rating for each outcome will be presented in a summary of findings table using the GRADEpro online tool (https://gradepro.org/). The review team will independently grade the overall quality of each outcome. Disagreements will be resolved through discussion or consensus.

## Sensitivity analysis
The impact of imputing missing data, potential confounders and studies with high risk of bias on the

robustness of the findings will be assessed using a sensitivity analysis, that is, the meta-analysis will be performed twice, with and without such studies.

## Patient and public involvement

The representatives of patients and public involvement groups were not involved in the development of this review protocol. The review findings will be disseminated to the public through social media and other public platforms.

## Ethics and dissemination

We will only retrieve and analyse data from existing studies, hence no ethical approval is required. The review findings will be published in a peer-reviewed journal and presented in scientific conferences.

**Acknowledgements** IA, KM and HD are supported by the NIHR Manchester Biomedical Research Centre. The first author is also supported by the Deanship of Scientific Research at the College of Applied Medical Sciences Research Center at King Saud University.

**Contributors** IA, the guarantor of this review, developed and drafted the initial version of this protocol and tested the preliminary search strategy. KM and HD contributed to the development, critically reviewed and provided extensive feedback on the earlier versions of this manuscript. Jan Schoones (an information specialist, Systematic Review Solutions) searched the databases. All authors approved this version of the manuscript.

**Funding** NIHR Manchester Biomedical Research Centre (funder reference: IS-BRC-1215-20007).

**Competing interests** None declared.

**Patient consent for publication** Not required.

**Provenance and peer review** Not commissioned; externally peer reviewed.

**ORCID iDs**
Ibrahim Almufarrij http://orcid.org/0000-0003-4043-7234
Harvey Dillon http://orcid.org/0000-0002-7036-7890
Kevin J Munro http://orcid.org/0000-0001-6543-9098

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
