## [Reviewer comments · BMJ Open]

ARTICLE DETAILS

TITLE (PROVISIONAL)	Is the outcome of fitting hearing aids to adults affected by whether an audiogram-based prescription formula is individually applied? A systematic review protocol.
AUTHORS	Almufarrij, Ibrahim; Dillon, Harvey; Munro, Kevin

VERSION 1 – REVIEW

REVIEWER	Enrique Lopez-Poveda University of Salamanca, Institute of Neuroscience of Castilla y Leon
REVIEW RETURNED	22-Dec-2020

GENERAL COMMENTS	This manuscript describes a protocol for a systematic review aimed at investigating whether hearing aids (or direct-to-consumer hearing devices) programmed using audiogram-based prescription rules result in better outcomes than hearing aids (or direct-to-consumer devices) programmed otherwise. The question is timely and relevant because direct-to-consumer hearing devices and over-the-counter hearing aids are becoming increasingly more popular and those devices, unlike conventional hearing aids, are not programmed based on the user's audiogram. Instead, they either come with a generic gain prescription for an average hearing loss, or can be adjusted by the user. The scope of the review, however, is ambiguous and some important aspects of the protocol and the manuscript are confusing or unclear. I am afraid that the protocol and manuscript would need to be substantially improved before I can recommend the manuscript for publication. Comments 1. I found it unclear whether the review will include only studies where participants were given the opportunity to compare the two approaches at different time instants (i.e., crossover designs) or will also include studies where different groups of participants were given different treatments but treatments were assessed using the same outcomes (i.e., between-group studies). The "Data synthesis" section in p. 8, suggest that the authors plan to include the two kinds of studies. If this is indeed the case, this should be acknowledged early on in the text, preferably in the Abstract. I am not convinced, however, that between-group studies can yield useful findings on any of the outcomes of interest, i.e., on the outcomes listed at the top of p. 6. Regarding the primary outcome (listening preference), how could anyone rate their preference on the type of intervention if they were only treated with one of them? In other words, how could anyone say that they prefer listening with devices programmed based on the audiogram if they were not given the opportunity to try those devices? Or the other way round, how could anyone say that they prefer listening with over-the-counter
--

	hearing aids if they were not given the opportunity to try them on? Similarly, it would be hard to assess the secondary outcomes (listed in p. 6, L6-14) using between-group studies, particularly if the groups are not well matched for age, hearing health status, and degree of hearing loss. Even the two interventions were assessed using common hearing tests or questionnaires, the scores would almost certainly be different for the two participant groups if the groups differed in age, hearing health, or degree of hearing loss. Maybe I am missing something, but based on my understanding of the manuscript, both the primary and secondary outcomes, can only be reasonably assessed using “cross-over” designs and it would not be possible to assess them using “between-group” designs unless the review restricts to between-group designs where groups are carefully matched for various critical variables, something not addressed in the manuscript. 2. It is also unclear if the review will include only studies that compare programming approaches using the same hearing device (as said in p. 3, L20-22), or will also include studies that compare programming approaches using different devices or even different types of devices, i.e., hearing aids versus direct-to-consumer hearing devices (as indicated in the Abstract, L7-10, and in the Rationale section of p. 4, L24-42). I am skeptical that a review of outcomes for devices that use different technologies would be meaningful or useful. The authors themselves acknowledge this issue in p. 3, L22-24 but do not convincingly address it. 3. It is also unclear if the review will include studies that are published (or accepted for publication), or will also include on-going studies. The abstract says both that the authors will only analyze data from published studies (p. 2, L40) and that the search will “identify published, completed but yet-to-be published and ongoing studies” (p. 2, L19-20). How could the results from unpublished or ongoing studies be trusted? 4. The information provided in the section Study Records is very generic and unspecific to the theme of the study. As it stands, the text in this section could apply to any systematic review of the literature in any field of study and indeed some passages appear to be copied and pasted from general texts. For example, in p. 9, L30-31, the authors talk about “dose-specific gradient”. What does this mean in the present context? Also, in p. 9, L31-32, they talk about “plausible confounding factors that likely have reduced the effect observed”. Which factors explicitly? The whole section entitled “Study Records” should be rewritten tailored to the specific theme and aims of the study. 5. The section entitled “Contributor” (p. 10) acknowledges “Jan Schoones (...) searched the databases”. Does this mean that the study has already been conducted? Altogether, I am sorry that I cannot recommend this study protocol for publication.
--	---

REVIEWER	Sarah Granberg Univ Orebro, School of health sciences
REVIEW RETURNED	16-Feb-2021

GENERAL COMMENTS	Thank you for the opportunity to review the current protocol. This is a fine review protocol addressing a very important area. The protocol is clear and the searches seem to have been adequately performed. The statistics is also clear and adequate given the methodology. I have only identified minor aspects that I would like the authors to elaborate a bit in their protocol. These aspects concern the intervention and the outcome. Intervention: In the rationale, the authors state that "numerous direct-to-consumer hearing devices have been mass-produced and marketed to customers. Although these hearing devices vary considerably in quality, a few have comparable electroacoustics to conventional hearing aids that are programmed and fitted by audiologists ". I can surely recognize that there might be many direct-to-consumer devices of various quality. My concern: how do the authors ensure that the included devices (the intervention) have comparable electroacoustics to conventional fitted HA:s (given that almost anyone who has HL and tries some sort of amplified device would experience some benefit)? To make a true comparison between the "intervention" and the "comparison", this matter needs to be attended to. Outcome: the authors state that the primary outcome is "listening preference". How do the authors anticipate that this aspect will be measured in the included studies? This matter also need to be addressed in the protocol. And lastly, a tip from me: there is a free web-tool designed to help researchers working with systematic reviews. It can be used during the entire process, e.g. when assessing data and extracting data (https://rayyan.qcri.org/welcome).
--

REVIEWER	Derek Hoare University of Nottingham, NIHR Nottingham Hearing Biomedical Research Unit
REVIEW RETURNED	20-Feb-2021

GENERAL COMMENTS	Thank you for the opportunity to review this protocol for a systematic review comparing outcomes from audiogram-based hearing aid prescription to other methods of fitting. The review is pre-registered on PROSPERO and uses standardised tools for appraising the evidence. The protocol provides an interesting concise history hearing aid prescribing, and the review is likely to make a valuable contribution to the field. I have only minor comments for completeness and making it align with the current registration in PROSPORO. Given the non-audiogram approach is described as the intervention the question and title of the review should reflect this. The current title and question suggest the audiogram-based approach is the intervention being tested. Alternatively, the title and questions could just describe the comparison. In terms of eligibility, will reporting any outcome of interest mean inclusion or must the primary outcome be reported? Planned databases are discrepant between the PROSPORO registration and the protocol; MEDLINE, GreyLit, OpenGrey, CINAHL... Presumably you had originally planned to include grey literature? But were other changes made to the plan after searches were undertaken? Discrepancy needs explanation. In data analysis please described how different studies designs
---

	(RCT, NRCT, crossover) will be handled. Presumably RCT and NRCTs will be synthesised separately. And crossover studies; will only first phase be included, and if not how will any potential carry over effects be considered?
--	--

VERSION 1 – AUTHOR RESPONSE

Reviewer 1	Responses
This manuscript describes a protocol for a systematic review aimed at investigating whether hearing aids (or direct-to-consumer hearing devices) programmed using audiogram-based prescription rules result in better outcomes than hearing aids (or direct-to-consumer devices) programmed otherwise. The question is timely and relevant because direct-to-consumer hearing devices and over-the-counter hearing aids are becoming increasingly more popular and those devices, unlike conventional hearing aids, are not programmed based on the user’s audiogram. Instead, they either come with a generic gain prescription for an average hearing loss, or can be adjusted by the user. The scope of the review, however, is ambiguous and some important aspects of the protocol and the manuscript are confusing or unclear. I am afraid that the protocol and manuscript would need to be substantially improved before I can recommend the manuscript for publication.	We thank the reviewer for the careful reading of the manuscript and the constructive comments. Our point-by-point responses to the reviewer’s comments and concerns are shown in the boxes below.
1. I found it unclear whether the review will include only studies where participants were given the opportunity to compare the two approaches at different time instants (i.e., crossover designs) or will also include studies where different groups of participants were given different treatments but treatments were assessed using the same outcomes (i.e., between-group studies). The “Data synthesis” section in p. 8, suggest that the authors plan to include the two kinds of studies. If this is indeed the case, this should be acknowledged early on in the text, preferably in the Abstract. I am not convinced, however, that between-group studies can yield useful findings on any of the outcomes of interest, i.e., on the outcomes listed at the top of p. 6. Regarding the primary outcome (listening preference), how could anyone rate their preference on the type of intervention if they were only treated with one of them? In other	We agree that the primary outcome of listening preference requires a within-group design, and that this is the most suitable design for comparing the two prescription approaches. However, secondary outcomes can be measured using within- and between-group design studies. As the review team expects to identify a limited number of studies, we will include within- and between-group designs. We have clarified this point in the ‘Strengths and limitations’ section. The risk of bias and imprecision in all these studies will be thoroughly examined, as described in the Method section (page 7–8, lines 24–32 and 1–4; pages 9–10, lines 16–31 and 1–6).

words, how could anyone say that they prefer listening with devices programmed based on the audiogram if they were not given the opportunity to try those devices? Or the other way round, how could anyone say that they prefer listening with over-the-counter hearing aids if they were not given the opportunity to try them on? Similarly, it would be hard to assess the secondary outcomes (listed in p. 6, L6-14) using between-group studies, particularly if the groups are not well matched for age, hearing health status, and degree of hearing loss. Even the two interventions were assessed using common hearing tests or questionnaires, the scores would almost certainly be different for the two participant groups if the groups differed in age, hearing health, or degree of hearing loss. Maybe I am missing something, but based on my understanding of the manuscript, both the primary and secondary outcomes, can only be reasonably assessed using “cross-over” designs and it would not be possible to assess them using “between-group” designs unless the review restricts to between-group designs where groups are carefully matched for various critical variables, something not addressed in the manuscript.	
2. It is also unclear if the review will include only studies that compare programming approaches using the same hearing device (as said in p. 3, L20-22), or will also include studies that compare programming approaches using different devices or even different types of devices, i.e., hearing aids versus direct-to-consumer hearing devices (as indicated in the Abstract, L7-10, and in the Rationale section of p. 4, L24-42). I am skeptical that a review of outcomes for devices that use different technologies would be meaningful or useful. The authors themselves acknowledge this issue in p. 3, L22-24 but do not convincingly address it.	We agree. As mentioned in the ‘Strengths and limitations section’, we will include only those studies that ‘compare programming approaches using the same hearing device. This information has been added to the Eligibility section (page 05, lines 25–27). We also broadened the inclusion criteria to include studies that used the same simulated hearing aid for both programming approaches. The manuscript has been modified accordingly (page 05, line 23). The registered protocol will be updated when this modification is accepted.
3. It is also unclear if the review will include studies that are published (or accepted for publication), or will also include on-going studies. The abstract says both that the authors will only	As mentioned in the Objective section, this review aims to collect and analyse all relevant published evidence. When appropriately conducted, systematic reviews can take anywhere from a few

analyze data from published studies (p. 2, L40) and that the search will “identify published, completed but yet-to-be published and ongoing studies” (p. 2, L19-20). How could the results from unpublished or ongoing studies be trusted?	weeks to years. Therefore, searching for ongoing studies or those completed but not yet published would enable the review team to identify studies that may have been published during the review process. The review team will contact the authors of these protocols to enquire about the studies’ publication status. To clarify, this information has been added to the revised manuscript as follows (page 07, lines 13–15). ‘The authors of all relevant identified protocols will be asked about the publication status of their clinical experiments to identify and include all studies published up until the review is completed.’ Furthermore, some databases, including PubMed, now search preprint servers. The use of filter functions to exclude preprint publications or trial protocols could omit some relevant citations by excluding non-indexed studies or those not yet indexed.
4. The information provided in the section Study Records is very generic and unspecific to the theme of the study. As it stands, the text in this section could apply to any systematic review of the literature in any field of study and indeed some passages appear to be copied and pasted from general texts. For example, in p. 9, L30-31, the authors talk about “dose-specific gradient”. What does this mean in the present context? Also, in p. 9, L31-32, they talk about “plausible confounding factors that likely have reduced the effect observed”. Which factors explicitly? The whole section entitled “Study Records” should be rewritten tailored to the specific theme and aims of the study.	We understand the difficulty in relating these descriptions to this specific review. As mentioned in the instructions to review authors, the authors of systematic reviews are strongly encouraged to follow the preferred reporting items for systematic review and meta-analysis protocols (PRISMA-P; Moher et al., 2015), which would facilitate the process of repeating, updating or critically evaluating the content of systematic reviews. If two review protocols followed the same guideline to build their Method sections, they should probably have similar structures, headings and subheadings and some standard content. The examples you mentioned are standard rules obtained from the grading of recommendations assessment, development and evaluation (GRADE) handbook. Changing the names of these rules could introduce some confusion. Therefore, in the revised manuscript, we added an example beside each rule as follows (page 09–10, lines 27–31 and 1).

	‘Notably, three factors may upgrade the quality of evidence: (i) large effect size, (ii) dose-response gradient (e.g., the magnitude of the participants’ preference for the audiogram-based approach [comparator] increased when the deviation from the prescribed target decreased) and (iii) plausible confounding factors that likely have reduced the effect observed (e.g., participants preferred the amplification characteristics for audiogram-based approach [comparator] over the other approaches [intervention] even when more advanced features were exclusively activated with the intervention).’
5. The section entitled “Contributor” (p. 10) acknowledges “Jan Schoones (...) searched the databases”. Does this mean that the study has already been conducted? Altogether, I am sorry that I cannot recommend this study protocol for publication.	This review is ongoing, and the screening process has not yet been completed. The databases were searched by an information scientist on 23 August 2020. To ensure that this review includes all up-to-date evidence, the searches will be repeated at the final write-up stage. This information has been added to the revised manuscript (page 02, lines 8–12; page 06, lines 15–20). We are sorry to hear this, and we hope that our clarification and revised manuscript will reassure the reviewer.
Reviewer 2	Responses
Thank you for the opportunity to review the current protocol. This is a fine review protocol addressing a very important area. The protocol is clear and the searches seem to have been adequately performed. The statistics is also clear and adequate given the methodology. I have only identified minor aspects that I would like the authors to elaborate a bit in their protocol. These aspects concern the intervention and the outcome.	We thank the reviewer for the careful reading of the manuscript and the constructive comments. Our point-by-point responses to the reviewer’s comments and concerns are shown in the boxes below.
Intervention: In the rationale, the authors state that “numerous direct-to-consumer hearing devices have been mass-produced and marketed to customers. Although these hearing devices vary considerably in quality, a few have comparable electroacoustics to conventional hearing aids that are programmed and fitted by audiologists “. I can surely recognize that there might be many direct-to-consumer devices of various quality. My concern: how do the authors	We agree. As mentioned in the ‘Strengths and limitations section’, we will include only those studies that ‘compare programming approaches using the same hearing device. This information has been added to the Eligibility section (page 05, lines 25–27). We also broadened the inclusion criteria to include studies that used the same simulated

ensure that the included devices (the intervention) have comparable electroacoustics to conventional fitted HA:s (given that almost anyone who has HL and tries some sort of amplified device would experience some benefit)? To make a true comparison between the “intervention” and the “comparison”, this matter needs to be attended to.	hearing aid for both programming approaches. The manuscript has been modified accordingly (page 05, line 23). The registered protocol will be updated when this modification is accepted.
Outcome: the authors state that the primary outcome is “listening preference”. How do the authors anticipate that this aspect will be measured in the included studies? This matter also need to be addressed in the protocol.	We have now stated that preference can be measured only in cross-over designs. Preference can be measured using force choice (i.e., do you prefer programme A over programme B or vice versa) and magnitude estimation e.g., programme A is much better than programme B, programme A is better than programme B, no difference, programme B is better than programme A, or programme B is much better than programme A. In the analysis, the difference in proportion between those preferring each condition and its 95% confidence interval (CI) will be computed. This information has been added to the revised manuscript as follows (page 08, lines 13–16). ‘For the primary outcome of interest, the participants’ listening preference, however, the difference in proportion between those preferring each condition and its 95% CI will be computed.’
And lastly, a tip from me: there is a free web-tool designed to help researchers working with systematic reviews. It can be used during the entire process, e.g. when assessing data and extracting data (https://rayyan.qcri.org/welcome).	Thank you so much for your suggestion. Indeed, the Rayyan platform is a valuable free tool for authors of systematic reviews. Failure to synchronise the offline work to the Rayyan servers may lead to data loss. The University of Manchester’s IT team will be consulted on the tool’s data safety and reliability.
Reviewer 2	Responses
Thank you for the opportunity to review this protocol for a systematic review comparing outcomes from audiogram-based hearing aid prescription to other methods of fitting. The review is pre-registered on PROSPERO and uses standardised tools for appraising the evidence. The protocol provides an interesting concise history hearing aid prescribing, and the review is likely to make a valuable contribution to	We thank the reviewer for the careful reading of the manuscript and the constructive comments. Our point-by-point responses to the reviewer’s comments and concerns are shown in the boxes below.

the field. I have only minor comments for completeness and making it align with the current registration in PROSPORO.	
Given the non-audiogram approach is described as the intervention the question and title of the review should reflect this. The current title and question suggest the audiogram-based approach is the intervention being tested. Alternatively, the title and questions could just describe the comparison.	Thank you for your suggestion. We have not substantially changed the title because we are not hypothesising that one approach is better than the other. Indeed, nothing in our review or conclusions will change if we refer to the audiogram-based approach as an intervention or comparator. The new title reads as follows: 'Does an individually applied audiogram-based adult prescription formula affect outcome? A systematic review protocol.'
In terms of eligibility, will reporting any outcome of interest mean inclusion or must the primary outcome be reported?	Eligible studies that used either the primary or secondary outcomes or both will be included. To clarify this point, the manuscript has been modified as follows (page 06, line 06–07). 'Studies reporting any of the above outcomes will be included'
Planned databases are discrepant between the PROSPORO registration and the protocol; MEDLINE, Greylit, OpenGrey, CINAHL... Presumably you had originally planned to include grey literature? But were other changes made to the plan after searches were undertaken? Discrepancy needs explanation.	As you mentioned, the initial plan was to include grey literature in the information sources. However, there is no agreed method of systematically identifying such studies. The likelihood of identifying grey literature studies from the resources mentioned in the registered protocol (i.e., Greylit and OpenGrey) is extremely low. Therefore, these two grey literature resources will be removed from the registered protocol when the revision is accepted.
In data analysis please described how different studies designs (RCT, NRCT, crossover) will be handled. Presumably RCT and NRCTs will be synthesised separately. And crossover studies; will only first phase be included, and if not how will any potential carry over effects be considered?	Data from studies that have different designs will be synthesised separately if the studies do not meet the requirements detailed in Morris and DeShon (2002). Data from studies that have different designs will be combined only if their effect sizes can be transferred to the same metric (e.g., standardised mean difference) and their effect sizes estimate the same treatment effect. This information has been added to the revised manuscript (page 08, lines 27–29). In crossover design studies, data from all phases will be included. We consider that crossover designs are the most appropriate design to assess preference. If the conditions were not counterbalanced in a study, this would be a

	serious limitation because carryover effects would not be controlled, which would be reflected in the quality we assign to that study.
--	--

VERSION 2 – REVIEW

REVIEWER	Sarah Granberg Univ Orebro, School of health sciences
REVIEW RETURNED	01-Apr-2021

GENERAL COMMENTS	Thank you, from my point of view, the reviewers' comments have been adequately addressed by the authors.
--

REVIEWER	Derek Hoare University of Nottingham, NIHR Nottingham Hearing Biomedical Research Unit
REVIEW RETURNED	06-Apr-2021

GENERAL COMMENTS	I thank the authors for response to reviewer comments. However, the manuscript is still discrepant with its registration on PROSPORO, and this should be explained in the protocol manuscript given searches are already done, and screening is already taking place. I am concerned by any changes to the proposed inclusion/exclusion based on reviewer comments unless they are accompanied by clear explanation in the protocol, sufficient to indicate that no change the search strategy would need to be made. Introducing a change to inclusion after screening has started needs strong justification in the protocol manuscript to clarify the decision was not based on screening findings. I would question the decision to exclude studies using different devices as intervention and comparator given interest in preferences and the authors expectation there will be only a small number of included studies. I would consider having a fuller review, particularly if that were the thought prospectively; include these studies and address the potential issue statistically; analyse separately, or random effects meta-analysis, and/or sensitivity analyses. Given the stage the review is at the manuscript should also explain when and why the decision was made to exclude grey literature searches. This is even more an issue now if planning to follow up published protocols to confirm whether the related trial has since been published. Drawing on published protocols also presents a potential selection bias towards RCTs which will more likely have a published protocol. If doing this why not also extend to searching trail registers and grey literature? I still feel the title is not explanatory of the proposed review given it refers to one approach and the outcome to that approach, rather than a comparison. I suggest dropping 'individually applied audiogram-based' and just refer to prescription formulae as being generic to all approaches.
--

VERSION 2 – AUTHOR RESPONSE

	Reviewer 2	Responses
1.	Thank you, from my point of view, the reviewers' comments have been adequately addressed by the authors.	Thank you.

	Reviewer 3	Responses
1.	I thank the authors for response to reviewer comments.	Thank you.
2.	However, the manuscript is still discrepant with its registration on PROSPORO, and this should be explained in the protocol manuscript given searches are already done, and screening is already taking place. I am concerned by any changes to the proposed inclusion/exclusion based on reviewer comments unless they are accompanied by clear explanation in the protocol, sufficient to indicate that no change the search strategy would need to be made. Introducing a change to inclusion after screening has started needs strong justification in the protocol manuscript to clarify the decision was not based on screening findings.	The two changes between the manuscript and the registered protocol are the: (i) exclusion of grey literature and (ii) inclusion of simulated hearing aids. The reasons for excluding the former from the information sources can be found below (see response 4). The initial plan for the latter was to exclude studies that used simulated hearing aids because they are less likely to reflect real-life benefit. However, to maximise the number of included studies, we broadened the inclusion criteria to include simulated hearing aids. Such a change has no direct impact on the search strategy or initial screening process because studies using conventional hearing aids or simulated hearing aids are usually indistinguishable based on the abstracts or the titles. Therefore, all potentially relevant studies that used simulated hearing aids made it to the full-text assessment stage. As it happens, none of the studies with simulated hearing aids were eligible for inclusion for various reasons, including the lack of probe-tube verification. The following has been added to the revised manuscript to clarify this point (page 5, lines 27–29), and similar clarification will be added to the manuscript with the review. “The initial plan was to exclude studies that used simulated hearing aids because they are less likely to reflect real-life benefit. However, to maximise the number of included studies, we broadened the inclusion criteria to include simulated hearing aids.”
3.	I would question the decision to exclude studies using different devices as intervention and comparator given interest in preferences and the authors expectation there will be only a small number of included studies. I would consider having a fuller review, particularly if that were the thought prospectively; include these studies and address the potential issue statistically; analyse separately, or random effects meta-analysis, and/or sensitivity analyses.	Thank you for your suggestion. Although this review may benefit from including studies that used different hearing devices for the intervention and comparator, this could introduce bias because different hearing devices (with different signal-processing strategies) may lead to different outcomes, even if they were set for the same hearing loss and prescription. The comments received earlier from reviewers 1 and 2 were consistent with the views of the review team, as they stressed the importance of including studies that used the same hearing device for both the intervention and the comparator. The same restriction was also implemented in the previously published review on the effectiveness of audiogram-based prescriptions (Mueller, 2005). Therefore, the review team is inclined to only include studies that used the same devices for both the intervention and the comparator.

4.	Given the stage the review is at the manuscript should also explain when and why the decision was made to exclude grey literature searches.	The initial plan was to include grey literature in the information sources, but they were excluded during the search stage because a preliminary search of GreyLit and OpenGrey found no relevant records. Searching hearing-related magazines, including Audacity, Hearing Review and ENT and Audiology News, is probably the most efficient way to find grey literature studies in audiology. However, there is no agreed method of efficiently and systematically searching these sources. Therefore, the review team is inclined to restrict the eligibility criteria to published peer-reviewed studies. This level of detail has now been added to the revised manuscript (page 6, lines 27–30) and will be explained in the manuscript of the review: “The initial plan was to include grey literature in the information sources, but they were excluded because a preliminary search produced no relevant records, and there is no agreed method of systematically searching such literature.”
5.	This is even more an issue now if planning to follow up published protocols to confirm whether the related trial has since been published. Drawing on published protocols also presents a potential selection bias towards RCTs which will more likely have a published protocol. If doing this why not also extend to searching trial registers and grey literature?	The review team is not concerned because, as mentioned in the revised manuscript, we will repeat all searches at the final write-up stage. Therefore, recently published trials (randomised or otherwise) will be identified and included in the review, regardless of whether they have published protocols.
6.	I still feel the title is not explanatory of the proposed review given it refers to one approach and the outcome to that approach, rather than a comparison. I suggest dropping ‘individually applied audiogram-based’ and just refer to prescription formulae as being generic to all approaches.	Thank you. To capture both the intervention and comparator, the title has been modified to read as follows: “Is the outcome of fitting hearing aids to adults affected by whether an audiogram-based prescription formula is individually applied? A systematic review protocol.”

VERSION 3 – REVIEW

REVIEWER	Derek Hoare University of Nottingham, NIHR Nottingham Hearing Biomedical Research Unit
REVIEW RETURNED	17-Jun-2021
GENERAL COMMENTS	Thank you for addressing my concerns. I have no further comments on the manuscript.